# Association of Metabolic Parameter Variability with Esophageal Cancer Risk: A Nationwide Population-Based Study

**DOI:** 10.3390/jpm12030375

**Published:** 2022-03-01

**Authors:** Ji Eun Lee, Kyungdo Han, Juhwan Yoo, Yohwan Yeo, In Young Cho, Belong Cho, Hyuktae Kwon, Dong Wook Shin, Jong Ho Cho, Yong-Moon Park

**Affiliations:** 1Department of Family Medicine, Seoul National University Hospital, Seoul 03080, Korea; 83342@snuh.org (J.E.L.); belong@snuh.org (B.C.); hyuktae@gmail.com (H.K.); 2Department of Statistics and Actuarial Science, Soongsil University, Seoul 06978, Korea; hkd@ssu.ac.kr; 3Department of Biomedicine & Health Science, The Catholic University of Korea, Seoul 06591, Korea; dbwnghks7@catholic.ac.kr; 4Department of Family Medicine/Supportive Care Center, Samsung Medical Center, School of Medicine, Sungkyunkwan University, Seoul 06351, Korea; yohwan.yeo@samsung.com; 5Department of Family Medicine, Kangbuk Samsung Hospital, Seoul 03181, Korea; inyoungs.cho@samsung.com; 6Department of Clinical Research Design & Evaluation, Samsung Advanced Institute for Health Science & Technology (SAIHST), Sungkyunkwan University, Seoul 06355, Korea; 7Department of Thoracic and Cardiovascular Surgery, Samsung Medical Center, School of Medicine, Sungkyunkwan University, Seoul 06351, Korea; 8Department of Epidemiology, Fay W. Boozman College of Public Health, University of Arkansas for Medical Sciences, Little Rock, AR 72205, USA; ypark@uams.edu

**Keywords:** claims data, esophageal cancer, metabolic parameter, nationwide study, variability independent of the mean

## Abstract

**Introduction:** Certain metabolic parameters increase the risk of esophageal cancer. This study investigated the association between the variability in metabolic parameters and esophageal cancer incidence using large nationally representative data. **Methods:** Using the health checkup and claims data provided by the Korean National Health Insurance Service (NHIS), we included 8,376,233 subjects who underwent NHIS-provided health checkups between 2009 and 2010 (index year) and two or more health checkups within five years before the index year. Hazard ratios (HRs) and 95% confidence intervals (CIs) for esophageal cancer were obtained using Cox proportional hazards models according to the quartiles of variability of each metabolic parameter: fasting blood glucose (FBG), weight, systolic blood pressure (SBP), and total cholesterol (TC) as well as a cumulative number of high-variability parameters. **Results:** A total of 6,455 cases of esophageal cancer occurred during a mean (±SD) follow-up of 8.8 (±1.1) years. The following metabolic parameters were used, with an adjusted HR and 95% CI: FBG (1.11, 1.03–1.18), weight (1.15, 1.07–1.23), SBP (1.08, 1.01–1.16), and TC (1.23, 1.15–1.32). The risk of esophageal cancer was higher in the highest quartile of variability than the lower quartiles. The risk of esophageal cancer gradually increased with a greater number of high-variability parameters: 1.08 (1.02–1.15), 1.22 (1.14–1.31), and 1.33 (1.21–1.46) for 1, 2, and 3–4 high-variability parameters (vs. none). **Conclusions:** A high variability of metabolic parameters was associated with an increased esophageal cancer risk. Further studies are needed to replicate our findings in other populations.

## 1. Introduction

Esophageal cancer is the seventh most common cancer worldwide and the sixth common cause of cancer-related death [1]. There are more than 600,000 new cases of esophageal cancer diagnosed annually and over 540,000 mortalities each year [1]. Specifically, Eastern Asia has the highest incidence and mortality rates of the disease [2]. Although the overall incidence of esophageal cancer has been decreasing worldwide, this has been increasing in some regions in North America and Europe [2]. Despite the improvement in treatment outcomes of esophageal cancer, the survival rates remain low. In fact, the 5-year survival rate in Korea was below 40% in 2009–2013 [3]. Thus, esophageal cancer is an important public health problem with a high incidence and low survival rate.

Many efforts are being made to identify the risk factors for the development of esophageal cancer. In particular, it has been found that certain metabolic parameters are associated with an increased risk of esophageal cancer [4]. For instance, obesity may increase the risk of esophageal cancer, especially esophageal adenocarcinoma (EAC) [5] while abdominal obesity may increase the risk of esophageal cancer independent of body mass index (BMI) [6]. Hypertension [7] and diabetes mellitus [8] were also associated with an increased risk of esophageal cancer. A recent systematic review showed that metabolic syndrome was associated with a risk of EAC [9].

Meanwhile, the relationship between the variability in metabolic parameters and cancer has recently been attracting attention. A study showed that high variability in fasting blood glucose (FBG), systolic blood pressure (SBP), total cholesterol (TC), and body weight were each associated with a risk of lung cancer [10]. Furthermore, weight variability has been associated with an increased risk of several cancers, such as hepatocellular carcinoma [11] and prostate cancer [12].

However, there has been no study on the relationship between the variability of metabolic parameters and the risk of esophageal cancer. Thus, this study aimed to reveal the association between variability in metabolic parameters and the incidence of esophageal cancer using nationally representative data. 

## 2. Materials and Methods

### 2.1. Data Source and Study Population

We used the health checkup data and claims data provided by the Korean National Health Insurance Service (NHIS). The NHIS provides medical coverage and medical aid to 97% and 3% of the Korean population, respectively. The NHIS database includes data regarding qualification for insurance (i.e., age, sex, and income level), diagnosis codes following the International Classification of Disease 10th revision (ICD-10), and claims data submitted by healthcare providers [13]. The NHIS also provides regular health checkups, including examinations on cardiovascular risk factors for all insured employees regardless of age and for those over 40 years of age every two years [14]. They measure metabolic parameters, such as blood pressure, body height, weight, and waist circumference, as well as take a blood sample (collected after overnight fasting). Questionnaires on health behavior and past medical history are also recorded. The NHIS database has been used in many epidemiological studies, the details of which can be found elsewhere [13,14].

In this study, we included those who underwent NHIS-provided health checkups between 2009 and 2010 (index year) and two or more health checkups within five years before the index year. Of the 17,664,057 people who underwent health checkups in the index year, 8,915,753 received over three health checkups during the period described. We excluded those with missing data for the necessary variables (n = 372,137), those who were with diagnosed cancer before the index date (n = 150,147), those diagnosed with esophageal cancer (n = 707), and those who died (n = 16,529) within 1 year after the index date for lag time. Finally, the study population included 8,376,233 subjects (Figure 1). This study was approved by the Institutional Review Board of Samsung Medical Center (IRB File No. SMC 2021-11-003), and the need for informed consent was waived because we used deidentified data for our analysis.

FBG, weight, SBP, and TC were selected as the metabolic parameters, according to the previous studies [10,11,15]. Variability was defined as the intraindividual variability measured by variability independent of the mean (VIM) in the FBG, weight, SBP, and TC values from the health checkup data. The VIM was calculated using the equation 100 × standard deviation (SD)/mean^β^; β is the regression coefficient, which is the natural logarithm of the SD divided by the natural logarithm of the mean [16]. The VIM is a transformation of the coefficient of variation using a regression coefficient, which is defined independently of the mean value [17,18]. High variability was defined as the highest quartile (Q4) of each variability while low variability was defined as the lower quartiles (Q1–Q3) of each variability. The metabolic parameter variability index was defined as the cumulative number of high-variability (Q4) metabolic parameters.

### 2.2. Study Outcomes and Follow-Up

The primary end point of this study was the incidence of esophageal cancer. This was defined by a diagnosis of esophageal cancer with the esophageal cancer code (C15) registration on the national copayment program for critical illnesses. In Korea, when a person is diagnosed with a cancer, only a 5% copayment applies for the cancer workup and treatment (vs. 20–30% for other common diseases). Thus, virtually all cancer patients register on this national copayment reduction program. Therefore, the cancer incidence in Korea is rarely omitted from this claims database and is sufficiently reliable. We followed the study population from baseline to the date of the new diagnosis of esophageal cancer, death, or until 31 December 2019, whichever came first.

### 2.3. Covariates

Information about smoking, alcohol consumption, and physical activity was obtained from questionnaires administered at the index year health checkup. Alcohol consumption was divided into three levels: nondrinking, mild to moderate drinking (<30 g/day), and heavy drinking (≥30 g/day) [19]. Regular physical activity was defined as moderate physical activity for more than 30 min at least 5 times per week or strenuous physical activity performed for more than 20 min at least 3 times a week [20]. Income level was divided into quartiles, and subjects with medical aid (~3% of population) were combined with the lowest quartile for the analyses.

The diagnosis of diabetes was confirmed if subjects had at least one claim for the ICD-10 codes E10–14 during the index year and a prescription for antidiabetic medication or if FBG levels were ≥126 mg/dL at the health checkup. The diagnosis of hypertension was confirmed if subjects had at least one claim for the ICD-10 codes I10 or I11 per year and a prescription for antihypertensive medication or if the SBP ≥ 140 mmHg or diastolic blood pressure (DBP) ≥ 90 mmHg was measured at the health checkup. Lastly, the diagnosis of dyslipidemia was confirmed if subjects had at least one claim for the ICD-10 code E78 per year and a prescription for lipid-lowering medication or if the TC levels were ≥240 mg/dL at the health checkup.

### 2.4. Statistical Analysis

We used descriptive statistics for baseline characteristics of the study population. Subjects were divided into four groups based on the cumulative number of high-variability metabolic parameters (i.e., FBG, weight, SBP, and TC): 0, 1, 2, and 3–4.

The hazard ratios (HRs) and 95% confidence intervals (CIs) for esophageal cancer incidence were estimated using a Cox proportional hazards model in each baseline metabolic parameter (i.e., FBG, BMI, blood pressure, and TC). Along with the crude analysis (Model 1), Model 2 included age and sex. Furthermore, Model 3 included income level and health behaviors, such as smoking, alcohol consumption, and regular physical activity. In Model 4, the baseline FBG, BMI, SBP, and TC (excluding the main parameter for each analysis) were added.

Then, HRs and 95% CIs were calculated according to the variability of metabolic parameters and the cumulative number of high-variability metabolic parameters using the same serial multivariate adjustment. The incidence probability of esophageal cancer according to the variability in each metabolic parameter (i.e., FBG, weight, SBP, and TC) and cumulative number of high-variability (Q4) parameters (metabolic parameter variability index) was calculated using Kaplan–Meier curves, and the log-rank test was performed to examine differences among the groups.

We performed further analysis for subgroups stratified according to age, sex, BMI, smoking, alcohol consumption, and the presence of cardiometabolic comorbidities (i.e., presence of hypertension, diabetes, or dyslipidemia). All statistical analyses were performed using the SAS version 9.4 (SAS Institute Inc. Cary, NC, USA), and the *p* values provided are two-sided, with statistical significance set at 0.05.

## 3. Results

### 3.1. Baseline Characteristics of Study Population

The subjects’ age increased as the cumulative number of high-variability parameters increased. Females, nonsmokers, and nondrinkers were more likely to have high-variability parameters. Subjects with high-variability parameters tended to have less regular physical activity. Subjects with lower income tended to be in the high-variability parameters group. Those with diabetes mellitus, hypertension, dyslipidemia, and metabolic syndrome were also more likely to have high-variability parameters (Table 1).

### 3.2. Risk of Esophageal Cancer According to Selected Metabolic Parameters

Regarding baseline metabolic parameters, among those with high FBG and high blood pressure, the risk of esophageal cancer was higher even after the multivariable adjustment: aHR (95% CI) 1.18 (1.12–1.24) for high FBG and 1.24 (1.17–1.31) for high blood pressure. Those with a high BMI had a lower risk of esophageal cancer risk than those with a low BMI: 0.76 (0.72–0.80). There was no significant association between the baseline TC and esophageal cancer in this study (Table 2).

### 3.3. Risk of Esophageal Cancer According to Level of Each Metabolic Parameter Variability

For each metabolic parameter (FBG, weight, SBP, and TC), the risk of esophageal cancer was higher in those in higher VIM groups than those in lower quartiles. The aHRs (95% CIs) for the Q2, Q3, and Q4 groups were 1.02 (0.95–1.09), 1.05 (0.98–1.13), and 1.11 (1.03–1.18) for FBG (*p* = 0.002); 1.05 (0.98–1.13), 1.02 (0.95–1.10), and 1.15 (1.07–1.23) for weight (*p* = 0.0003); 1.04 (0.96–1.11), 1.09 (1.02–1.17), and 1.08 (1.01–1.16) for SBP (*p* = 0.0127); and 0.99 (0.92–1.06), 1.06 (0.98–1.13), and 1.23 (1.15–1.32) for TC (*p* < 0.0001), respectively.

The risk of esophageal cancer gradually increased with the cumulative number of high-variability (Q4) parameters (metabolic parameter variability index). The aHRs (95% CIs) for groups with 1, 2, and 3–4 high-variability parameters were 1.08 (1.02, 1.15), 1.22 (1.14, 1.31), and 1.33 (1.21, 1.46), respectively, compared to the reference group (i.e., those with zero high-variability metabolic parameters) (Table 3, Figure 2).

### 3.4. Stratified Analyses

In the subgroup analyses, according to age, sex, BMI, smoking, alcohol consumption, and the presence of cardiometabolic comorbidity, the risk of esophageal cancer increased alongside the number of high-variability parameters in all subgroups, except in women. Specifically, the risk of esophageal cancer was higher in the younger age group (<65 years) (aHR of ≥3 high variability parameters: 1.48 (1.30–1.68) vs. 1.31 (1.15–1.49), *p* < 0.001), in males (1.35 (1.23–1.49) vs. 1.06 (0.77–1.46), *p* = 0.007) and in those without cardiometabolic comorbidities (1.39 (1.19–1.63) vs. 1.28 (1.14–1.43), *p* = 0.026). No significant difference in association was found for BMI, smoking, and alcohol consumption (*p* > 0.05) (Figure 3).

## 4. Discussion

To our knowledge, this is the first study to investigate the relationship between metabolic variability and the occurrence of esophageal cancer. We found that high degrees of variability in FBG, weight, SBP, and TC were associated with an increased risk of esophageal cancer. Furthermore, those with a greater number of high-variability parameters also had an increased risk of esophageal cancer, suggesting a dose–response relationship between the cumulative number of high-variability metabolic parameters and esophageal cancer risk.

Regarding the relationship between the baseline metabolic parameters and the incidence of esophageal cancer, high FBG and high blood pressure were associated with a risk of esophageal cancer. Conversely, BMI had an inverse association with esophageal cancer whereas TC did not show a significant association. However, regarding the variability of metabolic parameters, high variability in all parameters (FBG, weight, SBP, and TC) was positively associated with esophageal cancer.

Hyperglycemia is a known risk factor of various cancers: breast, pancreas, endometrium, etc. [21]. Furthermore, a recent meta-analysis indicated that diabetes mellitus is associated with a risk of esophageal cancer [8]. The suggested mechanisms of hyperglycemia in the occurrence of cancer include DNA damage, impairment of DNA repair, dysregulation of tumor suppressors, and inflammation [21]. In diabetes mellitus, gastric hypomotility and aggravated gastroesophageal reflux are considered additional mechanisms for the occurrence of esophageal cancer [8]. Both hyperglycemia and glucose variability have been associated with various cancers, including hepatocellular carcinoma [15] and gastric cancer [22]. In one study that analyzed cancer by organ system, glucose variability was associated with cancers of the digestive, respiratory, and intrathoracic systems as well as of genital organs [23]. Oscillating glucose has more deleterious effects than a constant glucose level on endothelial function and oxidative stress [24], which may aggravate carcinogenesis. Growth hormone disturbance is also considered a mechanism that explains the effect of glycemic variability [23].

In this study, the baseline BMI showed an inverse association with esophageal cancer, which agrees with previous studies. Several studies have demonstrated a lower risk of esophageal squamous cell carcinoma (ESCC) in people with obesity, in contrast to EAC [25,26]. ESCC is the most common histologic type in Asian countries, including Korea (~90%) [3,27]. The mechanism for this inverse association is uncertain; however, this is suspected to be due to micronutrient deficiencies or malnutrition in underweight persons, which can aggravate the occurrence of cancer [27]. Conversely, weight variability was highly associated with esophageal cancer risk in this study. Weight variability has been associated with the future incidence of several cancers, such as hepatocellular carcinoma, [11] lung cancer, [10] and prostate cancer [12]. In weight fluctuation, alterations in adipose tissue may induce hypoxia, leptin secretion, and chronic inflammation [28], which aggravate carcinogenesis.

High blood pressure and high SBP variability were associated with a risk of esophageal cancer. Although there are only few studies on the relationship between blood pressure and esophageal cancer, the study results agree with those of previous studies. Recent studies have showed a positive association between high blood pressure and esophageal cancer risk [7,25]. Hypertension is related to the shortening of telomeres, which can lead to cellular complications and carcinogenesis [29]. Furthermore, it is suspected that some antihypertensive agents may promote carcinogenesis [30]. For esophageal cancer, gastroesophageal reflux disease, which is one of the risk factors of esophageal cancer, is more common in patients with hypertension [7]. Additionally, blood pressure variability may have an independent effect on cancer occurrence via oxidative stress [31], endothelial dysfunction [32], and inflammation [33] which are known mechanisms of carcinogenesis [34].

In this study, TC showed no significant association with esophageal cancer risk. A previous study showed that the association between TC and cancer risk differed according to types of cancer [35]. The risks were higher in colon, prostate, and testicular cancers but lower in stomach, liver, and hematopoietic cancers. Furthermore, in that study, TC was not significantly associated with esophageal cancer [35]. Future studies are needed to assess how the effect of TC varies depending on the type of cancer. Meanwhile, high TC variability was positively associated with the risk of esophageal cancer. In previous studies, lipid variability was associated with cancer risk, including multiple myeloma [36] and lung cancer [10]. The suggested mechanism for this is because cholesterol plays an important role in the cell membrane and can affect the cell signaling pathway [37]. In lipid variability, cholesterol level variations may influence gene expression in cancer cells [36].

This study showed a clear dose–response association between the variability of metabolic parameters and esophageal cancer risk. Many previous studies have revealed that each metabolic parameter increases the risk of cancer through mechanisms, such as endothelial dysfunction [34], cell signaling pathway dysregulation [21], and inflammation [21]. Additionally, recent studies have showed that metabolic parameter variability may affect carcinogenesis via oxidative stress [24,31], endothelial dysfunction [24,32], hormonal disturbance [23], and chronic inflammation [28,33]. The higher the variability of metabolic parameters, the more likely these mechanisms may increase the risk of esophageal cancer via the aforementioned mechanisms.

Although further research is needed, this study implies that efforts to lower the variability of metabolic parameters may be helpful for preventing esophageal cancer and other cancers. To lower BP variability, a careful selection of an antihypertensive drug can be helpful. From a meta-analysis, SBP variability was reduced using calcium channel blockers and non-loop diuretic drugs but not by angiotensin-converting enzyme inhibitors, angiotensin receptor blockers, or beta blockers [38]. For lowering lipid variability, high-dose statins can be used [39]. Among oral hypoglycemics, dipeptidyl-peptidase-4 (DPP-4) enzyme inhibitors decrease glucose variability [40]. Additionally, it is necessary to emphasize medication compliance as well as the choice of drug. Furthermore, we should continue to recommend not only weight loss but also weight maintenance. We can expect that efforts to lower the variability of metabolic parameters, along with the classic risk factors of esophageal cancer (i.e., smoking, drinking, and low consumption of vegetables), can reduce the incidence of esophageal cancer.

Although this study has strengths, such as the use of a large nationwide database, there are some limitations to be mentioned. First, this was an observational study, so the association may not be causal. To minimize the effects of reverse causality, we excluded those diagnosed with esophageal cancer and those who died within one year from the index date. Second, we did not have data regarding the histologic types of esophageal cancer. Third, there was no information regarding the intentionality of body weight change. Moreover, changes in diet or physical activity might have affected the parameters of metabolic variability. Finally, we used Korean data; thus, the results may not be generalizable to other populations.

## 5. Conclusions

In conclusion, the presence of high-variability metabolic parameters was associated with an increased risk of esophageal cancer. The number of high-variability parameters showed a dose-dependent association with the risk of esophageal cancer. Further studies are needed to replicate our findings in other populations.

## Figures and Tables

**Figure 1 jpm-12-00375-f001:**
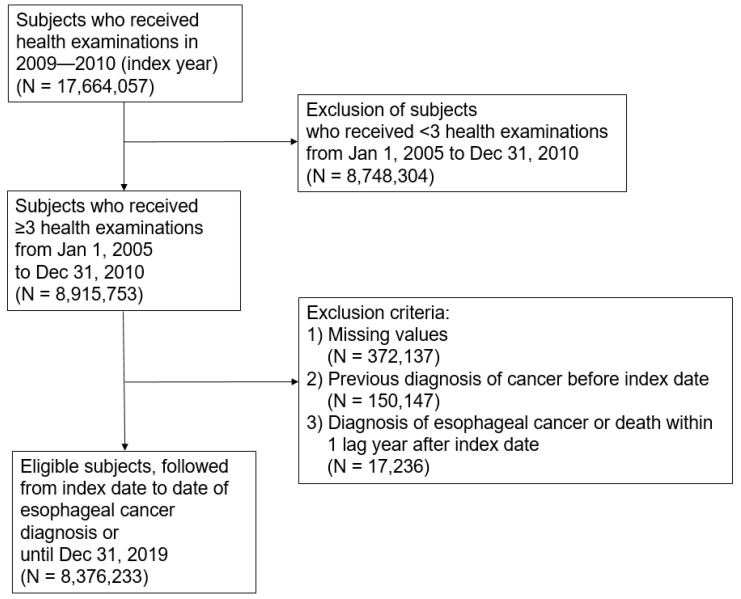
Flowchart of the study population.

**Figure 2 jpm-12-00375-f002:**
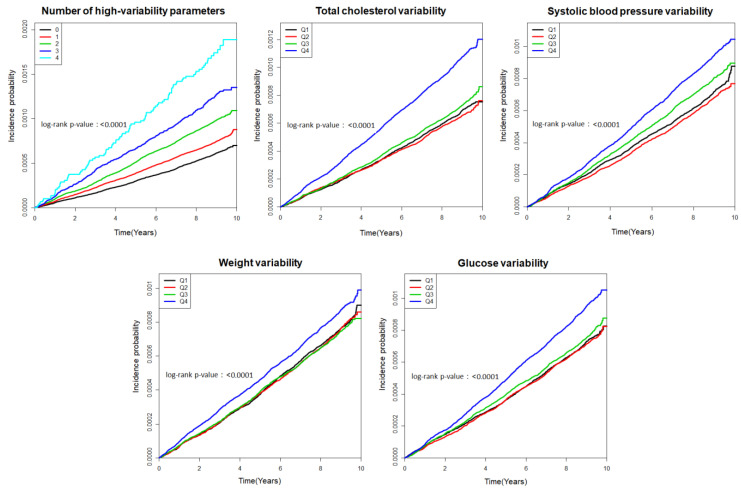
Kaplan−Meier curves of the cumulative incidence of esophageal cancer according to the number of high-variability parameters and quartiles of variability in each metabolic parameter.

**Figure 3 jpm-12-00375-f003:**
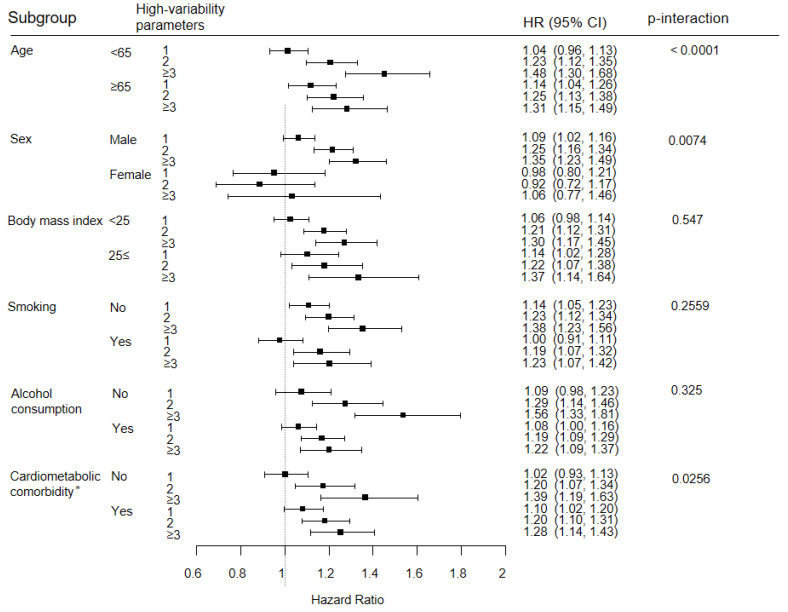
Stratified analyses. * Presence of cardiometabolic comorbidities (hypertension, diabetes, or dyslipidemia).

**Table 1 jpm-12-00375-t001:** Baseline characteristics of the study population by the metabolic parameter variability index.

	Metabolic Parameter Variability Index ^1^				
N	0	1	2	3,4	*p*-Value ^2^
	2,844,142	3,283,601	1,712,015	536,475	
Age (years)	47.7 ± 12.7	48.4 ± 13.8	49.6 ± 14.7	51.3 ± 15.7	<0.0001
Sex (male)	1,768,372 (62.2)	1,893,571 (57.7)	928,613 (54.2)	276,325 (51.5)	<0.0001
Smoking					<0.0001
Non-smoker	1,615,543 (56.8)	1,934,020 (58.9)	1,038,406 (60.7)	333,159 (62.1)	
Ex-smoker	507,319 (17.8)	524,227 (16.0)	255,203 (14.9)	77,095 (14.4)	
Current smoker	721,280 (25.4)	825,354 (25.1)	418,406 (24.4)	126,221 (23.5)	
Alcohol consumption					<0.0001
None	1,360,299 (47.8)	1,684,049 (51.3)	935,173 (54.6)	312,442 (58.2)	
Mild to moderate (<30 g/day)	1,262,834 (44.4)	1,350,945 (41.1)	649,196 (37.9)	184,053 (34.3)	
Heavy (≥30 g/day)	221,009 (7.8)	248,607 (7.6)	127,646 (7.5)	39,980 (7.5)	
Regular physical activity	575,148 (20.2)	638,725 (19.5)	319,768 (18.7)	94,591 (17.6)	<0.0001
Household income					<0.0001
Q1 + medical aid	507,532 (17.8)	652,828 (19.9)	367,788 (21.5)	120,566 (22.5)	
Q2	482,107 (17.0)	624,070 (19.0)	350,624 (20.5)	114,106 (21.3)	
Q3	761,070 (26.8)	917,239 (27.9)	482,447 (28.2)	151,350 (28.2)	
Q4	1,093,433 (38.5)	1,089,464 (33.2)	511,156 (29.9)	150,453 (28.0)	
Diabetes, yes	144,774 (5.1)	271,244 (8.3)	211,523 (12.4)	99,768 (18.6)	<0.0001
Hypertension, yes	654,110 (23.0)	910,482 (27.7)	562,981 (32.9)	210,196 (39.2)	<0.0001
Dyslipidemia, yes	423,202 (14.9)	604,694 (18.4)	386,067 (22.6)	147,318 (27.5)	<0.0001
Metabolic syndrome, yes	618,082 (21.7)	843,078 (25.7)	516,276 (30.2)	191,601 (35.7)	<0.0001
Weight (kg)	65.1 ± 11.2	64.3 ± 11.5	63.6 ± 11.8	62.8 ± 12.1	<0.0001
Height (cm)	165.1 ± 9.0	164.2 ± 9.2	163.2 ± 9.4	162.2 ± 9.5	<0.0001
Waist circumference (cm)	80.6 ± 8.7	80.6 ± 8.9	80.7 ± 9.1	80.9 ± 9.3	<0.0001
Body mass index (kg/m^2^)	23.8 ± 3.0	23.8 ± 3.1	23.8 ± 3.3	23.7 ± 3.4	<0.0001
Fasting blood glucose (mg/dL)	95.6 ± 17.4	96.9 ± 21.9	98.8 ± 26.7	101.6 ± 32.7	<0.0001
Systolic BP (mmHg)	122.7 ± 13.2	122.5 ± 14.7	122.6 ± 16.0	122.8 ± 17.6	<0.0001
Diastolic BP (mmHg)	76.7 ± 9.4	76.4 ± 9.8	76.3 ± 10.3	76.2 ± 10.9	<0.0001
Total cholesterol (mg/dL)	198.6 ± 33.6	195.8 ± 36.0	193.2 ± 38.9	190.6 ± 42.5	<0.0001
HDL cholesterol (mg/dL)	55.1 ± 22.7	55.4 ± 24.2	55.5 ± 25.5	55.6 ± 28.0	<0.0001
LDL cholesterol (mg/dL)	117.6 ± 34.9	114.6 ± 37.2	111.8 ± 39.5	109.0 ± 42.3	<0.0001
Triglycerides (geometric mean)	114.4 (114.4, 114.5)	113.3 (113.2, 113.4)	114.4 (114.3, 114.5)	115.6 (115.4, 115.8)	<0.0001
Glucose VIM	7.1 ± 3.1	9.9 ± 5.7	9.9 ± 5.7	15.8 ± 6.6	<0.0001
Weight VIM	1.3 ± 0.6	1.9 ± 1.3	2.5 ± 1.7	3.4 ± 1.9	<0.0001
Systolic BP VIM	7.0 ± 2.9	9.3 ± 5.0	11.5 ± 5.6	14.0 ± 5.4	<0.0001
Total cholesterol VIM	13.8 ± 5.8	19.0 ± 11.3	25.3 ± 13.8	32.5 ± 14.2	<0.0001

N, number; BP, blood pressure; VIM, variability independent of the mean. ^1^ Metabolic parameter variability index was defined as the cumulative number of high variability in each metabolic parameter (fasting blood glucose, body weight, systolic blood pressure, and total cholesterol levels). ^2^
*p*-values were calculated using chi square tests for categorical variables and the Student’s *t*-test or the Mann−Whitney U test for continuous variables.

**Table 2 jpm-12-00375-t002:** Risk of esophageal cancer according to selected metabolic parameters.

	N	Events (n)	Follow-up Duration (Person-Years)	Incidence Rate per 100,000	Model 1 HR ^1^	Model 2 aHR ^2^	Model 3 aHR ^3^	Model 4 aHR ^4^
**Fasting blood glucose (** **mg/dL)**								
<100	5,728,765	3368	50,851,612	6.62	1.00	1.00	1.00	1.00
≥100 or on meds ^5^	2,647,468	3087	23,136,523	13.34	2.02 (1.92, 2.12)	1.18 (1.12, 1.23)	1.13 (1.08, 1.19)	1.18 (1.12, 1.24)
**Body mass index (kg/m^2^)**								
<25	5,597,386	4648	49,389,063	9.41	1.00	1.00	1.00	1.00
≥25	2,778,847	1807	24,599,072	7.35	0.78 (0.74, 0.82)	0.75 (0.71, 0.79)	0.78 (0.73, 0.82)	0.76 (0.72, 0.80)
**Blood pressure (mmHg)**								
<130/85	4,534,843	1991	40,384,710	4.93	1.00	1.00	1.00	1.00
≥130/85 or on meds ^6^	3,841,390	4464	33,603,425	13.28	2.70 (2.56, 2.84)	1.19 (1.13, 1.26)	1.15 (1.09, 1.21)	1.24 (1.17, 1.31)
**Total cholesterol (** **mg/dL)**								
<240	7,427,173	5790	65,606,785	8.83	1.00	1.00	1.00	1.00
≥240 or on meds ^7^	949,060	665	8,381,350	7.93	0.90 (0.83, 0.97)	1.03 (0.95, 1.11)	1.01 (0.93, 1.10)	1.03 (0.95, 1.12)

N, number of subjects; n, number of esophageal cancer events; HR, hazard ratio; aHR, adjusted hazard ratio; Q, quartile; VIM, variability independent of the mean; ^1^ unadjusted; ^2^ adjusted for age, sex; ^3^ adjusted for age, sex, income level, smoking, alcohol consumption, and regular physical activity; ^4^ adjusted for variables in model 3, baseline fasting blood glucose, baseline body mass index, baseline systolic blood pressure, and baseline cholesterol; ^5^ fasting blood glucose ≥100 mg/dL and/or having been prescribed antidiabetic medication; ^6^ systolic blood pressure ≥130 mmHg, diastolic blood pressure ≥85 mmHg and/or having been prescribed antihypertensive medication; ^7^ total cholesterol ≥240 mg/dL and/or having been prescribed lipid-lowering medication.

**Table 3 jpm-12-00375-t003:** Esophageal cancer risk by quartiles of metabolic parameters variability and cumulative number of high variability in each metabolic parameter.

	N	Events (n)	Follow-up Duration (Person-Years)	Incidence Rate per 100,000	Model 1 HR ^1^	Model 2 aHR ^2^	Model 3 aHR ^3^	Model 4 aHR ^4^
**Glucose variability (VIM)**								
Q1	2,094,061	1477	18,424,532	8.02	1.00	1.00	1.00	1.00
Q2	2,094,172	1470	18,545,087	7.93	0.99 (0.92, 1.06)	1.04 (0.97, 1.12)	1.02 (0.95, 1.10)	1.02 (0.95, 1.09)
Q3	2,093,939	1568	18,568,704	8.44	1.05 (0.98, 1.13)	1.10 (1.03, 1.19)	1.06 (0.99, 1.14)	1.05 (0.98, 1.13)
Q4	2,094,061	1940	18,449,812	10.52	1.31 (1.22, 1.40)	1.21 (1.14, 1.30)	1.13 (1.06, 1.21)	1.11 (1.03, 1.18)
*p* for trend					<0.0001	<0.0001	0.0002	0.002
**Weight variability (VIM)**								
Q1	2,093,667	1576	18,516,003	8.51	1.00	1.00	1.00	1.00
Q2	2,092,118	1572	18,582,887	8.46	0.99 (0.93, 1.07)	1.09 (1.01, 1.17)	1.07 (1.00, 1.15)	1.05 (0.98, 1.13)
Q3	2,096,354	1540	18,577,380	8.29	0.97 (0.91, 1.04)	1.08 (1.01, 1.16)	1.06 (0.99, 1.13)	1.02 (0.95, 1.10)
Q4	2,094,094	1767	18,311,865	9.65	1.13 (1.06, 1.21)	1.27 (1.18, 1.36)	1.22 (1.14, 1.31)	1.15 (1.07, 1.23)
*p* for trend					0.0009	<0.0001	<0.0001	0.0003
**Systolic blood pressure variability (VIM)**								
Q1	2,094,065	1471	18,460,408	7.97	1.00	1.00	1.00	1.00
Q2	2,094,057	1408	18,610,077	7.57	0.95 (0.88, 1.02)	1.04 (0.97, 1.12)	1.04 (0.96, 1.12)	1.04 (0.96, 1.11)
Q3	2,094,074	1652	18,569,332	8.90	1.12 (1.04, 1.20)	1.11 (1.04, 1.20)	1.09 (1.02, 1.17)	1.09 (1.02, 1.17)
Q4	2,094,037	1924	18,348,318	10.49	1.32 (1.23, 1.41)	1.13 (1.06, 1.21)	1.09 (1.02, 1.17)	1.08 (1.01, 1.16)
*p* for trend					<0.0001	<0.0001	0.006	0.0127
**Total cholesterol variability (VIM)**								
Q1	2,094,058	1410	18,476,792	7.63	1.00	1.00	1.00	1.00
Q2	2,094,067	1358	18,630,457	7.29	0.95 (0.89, 1.03)	1.00 (0.93, 1.08)	0.99 (0.92, 1.07)	0.99 (0.92, 1.06)
Q3	2,094,050	1517	18,593,772	8.16	1.07 (0.99, 1.15)	1.09 (1.01, 1.17)	1.07 (1.00, 1.15)	1.06 (0.98, 1.13)
Q4	2,094,058	2170	18,287,114	11.87	1.56 (1.46, 1.66)	1.29 (1.21, 1.38)	1.26 (1.18, 1.35)	1.23 (1.15, 1.32)
*p* for trend					<0.0001	<0.0001	<0.0001	<0.0001
**Cumulative number of high-variability (Q4) in each parameter**								
0	2,844,142	1730	25,329,242	6.83	1.00	1.00	1.00	1.00
1	3,283,601	2417	29,041,630	8.32	1.22 (1.15, 1.30)	1.13 (1.07, 1.21)	1.10 (1.03, 1.17)	1.08 (1.02, 1.15)
2	1,712,015	1637	14,996,225	10.92	1.60 (1.50, 1.71)	1.33 (1.25, 1.43)	1.26 (1.18, 1.35)	1.22 (1.14, 1.31)
3,4	536,475	671	4,621,038	14.52	2.13 (1.95, 2.33)	1.53 (1.40, 1.67)	1.41 (1.29, 1.54)	1.33 (1.21, 1.46)
*p* for trend					<0.0001	<0.0001	<0.0001	<0.0001

N, number of subjects; n, number of esophageal cancer events; HR, hazard ratio; aHR, adjusted hazard ratio; Q, quartile; VIM, variability independent of the mean; ^1^ unadjusted; ^2^ adjusted for age, sex; ^3^ adjusted for age, sex, income level, smoking, alcohol consumption, and regular physical activity; ^4^ adjusted for variables in model 3, baseline fasting blood glucose, baseline body mass index, baseline systolic blood pressure, and baseline cholesterol.

## Data Availability

Restrictions apply to the availability of these data. Data was obtained from the Korean National Health Insurance Sharing Service and are available from the authors with the permission of the Korean National Health Insurance Sharing Service.

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
