# Peer review of "Association of Metabolic Parameter Variability with Esophageal Cancer Risk: A Nationwide Population-Based Study"

_jpm, 2022, doi:10.3390/jpm12030375_

Round 1

Reviewer 1 Report

The manuscript "Association of Metabolic Parameter Variability with Esophageal Cancer Risk: A Nationwide Population-Based Study" by Lee et al. reports the results of the study aimed to evaluate the relationship between baseline metabolic parameters (fasting blood glucose, weight, systolic blood pressure , and total cholesterol), as well as metabolic variability, and the occurrence of esophageal cancer. The manuscript is well written, providing information on the analysis of data on metabolic, lifestyle behavior and medical history of 8,376,233 subjects, obtained from Korean National Health Insurance Service. Results of this study point to the importance of loss of metabolic robustness, as evidenced by an increase in metabolic variability, in occurrence of esophageal cancer. The interesting finding of this study is the inverse association between weight and incidence of esophageal cancer. In line with proposed involvement of micronutrients deficiency in undernourished people, it would be interesting to evaluate, in the future, the contribution of alcohol consumption in that population as it often causes sub-clinical niacin deficiency which is known to have an impact on cancer risk.

The minor correction is needed in table 2.: "Fasting blood glucose" should be replaced by "Elevated fasting blood glucose" ( or "No", in the line bellow, replaced by "Not elevated").

Reviewer 2 Report

The presented manuscript closely follows the logic and structure of the paper published by the same authors regarding lung cancer. However, I do not see any considerable flaws in the statistical analysis and presentation of the results. The only thing is that weight and systolic blood pressure do not seem to me like metabolic parameters.

Author Response

Thank you for your comment. When we began this study, we selected the metabolic parameters from the definition of metabolic syndrome (3 or more of obesity, hyperglycemia, raised blood pressure, elevated triglyceride, and low HDL cholesterol). Because metabolic syndrome is known to increase risk of many types of cancer (Ex>Metabolic syndrome and risk of cancer: a systematic review and meta-analysis." Diabetes care 35.11 (2012): 2402-2411.). Also, recent studies suggest association between weight, high blood pressure and cancer incidence. (references of this manuscript 7,10,11,12,25, etc.). So, we included weigh and systolic blood pressure as the metabolic parameters
